# The Variability of Proximate Composition, Sugars, and Vitamin C in Natural, Organic, and Biodynamic, and Fermented Leaves of Fireweed (*Chamerion angustifolium* (L.) Holub (*Onagraceae*))

Marius Lasinskas [1,*], Elvyra Jariene [1], Nijole Vaitkeviciene [1], Jurgita Kulaitiene [1], Sonata Trumbeckaite [2], Aloyzas Velicka [1] and Ewelina Hallmann [3,4]

1   Department of Plant Biology and Food Sciences, Agriculture Academy, Vytautas Magnus University, Donelaicio St. 58, 44248 Kaunas, Lithuania; elvyra.jariene@vdu.lt (E.J.); nijole.vaitkeviciene@vdu.lt (N.V.); jurgita.kulaitiene@vdu.lt (J.K.); aloyzas.velicka@vdu.lt (A.V.)
2   Department of Pharmacognosy, Faculty of Pharmacy, Lithuanian University of Health Sciences, Sukileliu Av. 13, LT-50162 Kaunas, Lithuania; sonata.trumbeckaite@lsmu.lt
3   Department of Functional and Organic Food, Institute of Human Nutrition Sciences, Warsaw University of Life Sciences, Nowoursynowska 15c, 02-776 Warsaw, Poland; ewelina_hallmann@sggw.edu.pl or ewelina.hallmann@vdu.lt
4   Bioeconomy Research Institute, Agriculture Academy, Vytautas Magnus University, Donelaicio St. 58, 44248 Kaunas, Lithuania
*   Correspondence: marius.lasinskas@vdu.lt; Tel.: +370-676-82266

**Abstract:** Functional foods and herbs are becoming more and more popular as a way to improve health and at the same time improve diet. One of these plants is fireweed, which is abundant in fibers, proteins, and vitamin C in addition to polyphenols and carotenoids. Limited study is being carried out and there is limited information available about how the solid-phase fermentation and different growth systems change the proximate composition, as well as quantities of vitamin C, and sugars in the fireweeds leaves. The experiment was conducted in 2022 on an organic farm (Jonava district, Lithuania). The objective of this research was to determinate the impact of various growing systems (naturally, organically, and biodynamically) and duration (24, 48, and 72 h) effect of solid-phase aerobic fermentation on the changes of fibers, ash, proteins, vitamin C, and sugars. The fibers were established using the Kjeldahl method. High-performance liquid chromatography was used for sugars and vitamin C identification. The study found significant differences in the effects of various growing systems and solid-phase fermentation on the quantitative composition of substances in the fireweed leaves. Based on the data available, it is recommended to consume fermented fireweed leaves that are organically grown and fermented for a period of 48 h as they are a good source of vitamin C and fibers. Biodynamically grown fireweed leaves are a good source of ash and proteins. Naturally grown fireweed leaves, which are not fermented, are a good source of sugars.

**Keywords:** organic; biodynamic; fermentation; fireweed; vitamin C

## 1. Introduction

Researchers are looking for an alternative approach and there is a need for systems that can promote sustainable practices in agriculture as a result of growing worries about the proliferation of various chemical inputs in agricultural environments and their effects on the economy and society. Low-cost, organic, biodynamic, and biological farms are examples of alternative farming systems. Such farms provide sufficient quantities of high-quality food while being lucrative, socially and environmentally responsible, and safe for the environment [1]. Numerous studies have demonstrated that as compared to conventional farming, organic farming may improve soil biodiversity and biological activity [2]. It has also been claimed that biodynamic farming, a particular type of organic farming, maintains greater soil quality than conventional agricultural methods. Other than differing farming

practices, other variables that may affect biological soil features include plant soil type, species, and tillage [3–5].

According to some writers, organically or biodynamically treated soil contains more microorganisms than conventionally maintained soil. Microorganisms generate a variety of chemicals and increase the availability of those elements to plant roots. The soil microorganisms that are in the soil partly explain the higher mineral content and quality of organic crops [5,6].

Fireweed (*Chamerion angustifolium* (L.) Holub (*Onagraceae*) is a well-known medicinal plant used in traditional medicine globally. It grows in various soils, but usually in damaged areas: burned or cut forests and highways. It can grow in light forests, but not in full shade. It has a wide range of therapeutic effects: anti-proliferative, anti-inflammatory, antibacterial, and antioxidant. Flavonoids and ellagitaninns, such as oenothein B, are one of the most important biologically active compounds present in fireweed extracts. In traditional medicine, these herbs are used to treat benign prostatic hyperplasia, urinary problems, inflammation, pain, and spasms [7]. As a result, it is crucial to understand the chemical makeup and pharmacological characteristics of fireweeds. Nevertheless, there is not much extensive study available on this topic. To compare the impact of biodynamic and organic farming on fireweed leaf quality, a comparative study is needed.

One of the ways to improve extraction and modify bioactive compounds in fireweed fermented leaves and improve their bioavailability is solid-phase fermentation (SPF), when no additional water is added. It is expected that cutting and pressing during SPF could intensify cell wall degradation, thus improving the diffusion of bioactive compounds from the inner cell parts, and then initiating better extraction. In addition, there is evidence that some solid-phase fermentation parameters may activate the accumulation process of some bioactive compounds in fireweed leaves. Solid-phase fermentation technology can enhance fireweed tea and other products. Also during this process, microbes and enzymes are quite active inside the cells in addition to biochemical reactions [8].

It appears that the quantity of bioactive chemicals present in plants and their availability for infusion are crucial. Finding a method to make physiologically active chemicals in plants more bioavailable is crucial. Solid-phase fermentation is one of the modern methods for controlling biologically active substances and their bioavailability in the leaves of fireweed.

The use of the biodynamic system has not been established and an analysis of its effectiveness in organic systems has not yet been carried out. Additionally, no efforts have been made to examine how various farming methods affect the buildup of bioactive chemicals, and also proximate composition in fireweed leaves. To optimize the amounts of bioactive substances with health-promoting qualities, the optimal agronomic practices must be matched.

According to the findings of our study, farmers may choose a more profitable method of cultivating fireweeds utilizing biodynamic farming methods, and manufacturers could use fireweed leaf extracts in the production of high-quality dietary supplements.

## 2. Materials and Methods

### 2.1. Experiment

A field experiment was carried out on Giedres Nacevicienes organic farm in 2022, located in Safarkos village, Jonava district, Lithuania. The farm has been following an organic cultivation system for the past 10 years. The experimental plot covered an area of 2000 m$^2$ and consisted of perennial fireweeds that have been grown for the past four years. As part of the experiment, one section was left to allow fireweed to grow by itself naturally.

Fireweed plants were grown organically and biodynamically. The plants that were artificially grown were compared with the ones that were grown naturally, which were used as a control sample. The soil was fertilized with either organic compost (25 t ha$^{-1}$) (its composition: mineral nitrogen—52.73 mg kg$^{-1}$, available phosphorus—1932.49 mg kg$^{-1}$, and pH$_{KCl}$—6.97,) or biodynamic compost (25 t ha$^{-1}$) (its composition: mineral nitrogen—

51.09 mg kg$^{-1}$, available phosphorus—1591.08 mg kg$^{-1}$, and pH$_{KCl}$—6.83), two weeks before the beginning of plant vegetation (in the middle of June). Biodynamic experimental fields were treated with biodynamic (BD) preparation 500 in May (2nd decade) at a concentration of 1% solution (200 l/ha). BD preparation 500 is fermented manure that contains the following composition: total nitrogen—2.28%, available potassium—291 mg kg$^{-1}$, available phosphorus—1668 mg kg$^{-1}$, pH$_{KCl}$—6.86, saccharase enzyme activity—31.8 mg glucose 1 g$^{-1}$ soil per 48 h, and urease enzyme activity—1.64 mg NH$_3$ 1 g$^{-1}$ soil per 24 h. The leaves were sprayed twice: during the vegetation period with BD preparation 501 at a 0.5% solution—once in the morning during the stage of leaf formation in the middle of June, and the second time at the beginning of the plant's mass flowering in July (1st decade) (200 L/ha). BD preparation 501 is a finely powdered quartz crystal rock (SiO$_2$—99.8%). In the biodynamic system, the soil and plants of fireweed were treated with biodynamic preparations using the same methods as European biodynamic farms [9].

Organic experimental fields were sprayed with water in the middle of June and leaves of the fireweed were sprayed with water 2 times during the vegetation period—in the morning at the stage of leaf formation at the end of June and at the beginning of plant mass flowering (1 July decade).

Plant protection products against diseases and vermin were not used. Biodynamic compost and BD preparations 500 and 501 used in the experiment were purchased from a Demeter-certified farm (CvW KG, Internationale Biodynamische Praparatezentrale, Darmstadt, Germany).

The variants were rendered in three replications. From each replication, 10.8 kg of herbal raw material was collected for testing. Test boxes with and without biodynamic additives were established at the same sampling site (farm).

### 2.2. Plant Material and Solid-Phase Fermentation

At the start of mass blooming (1 July decade), the raw material was selected from three replications in each system. A total of 10.8 kg of leaves from each agricultural system was sampled.

- Control (naturally grown): 3.6 kg for control, 3.6 kg for fermentation 24 h, and 1.2 kg for 48 h.
- Fireweed grew organically: 3.6 kg for control, 3.6 kg for fermentation 24 h, and 3.6 kg for 48 h.
- Fireweed that has been grown biodynamically: 3.6 kg for control, 3.6 kg for fermentation 24 h, and 3.6 kg for 48 h.

Control (0 h): leaves that weren't fermented but were nonetheless kept for the intended time.

The raw material was cut with specialized plastic blades for the solid-phase fermentation and was separated into 3 samples, each weighing 1.2 kg. The material prepared was firmly packed into glass jars and sealed with an airtight lid. For 24 and 48 h, the fermentation process was conducted in the chamber at 30 °C. A ZIRBUS sublimation dryer $3 \times 4 \times 5$ (ZIRBUS Technology, Bad Grund, Germany) was used to lyophilize the raw materials after they had undergone fermentation and were frozen at $-35$ °C. To prepare the leaves for further analysis, they were ground to a powder (Grindomix GM 200 laboratory mill, Retsch GmbH, Haan, Germany).

### 2.3. Laboratory Analyses
#### 2.3.1. Chemicals and Reagents

ABTS$^{\bullet+}$ (2′2-azinebis-3-ethylbenzothiazolin-6-sulfonic acid), acetone (HPLC grade); acetic acid (glacial, 99.9% purity), deionized water; ethanol (HPLC grade), meta-phosphoric acid (HPLC grade), sodium acetate (99.9% purity), sugar standards (glucose, fructose, and sucrose); all standards with HPLC purity, vitamin C standards: dehydroascorbic acid and l-ascorbic acid; all standards with HPLC purity.

### 2.3.2. Sugars Identification and Quantification

One hundred milligrams of fireweed material was weighed. Five milliliters of 80% acetone was added and mixed with a vortex [10]. The mixture was then extracted in an ultrasonic bath for 10 min at 30 °C and 5.5 kHz. The samples were centrifuged for 10 min at 6,000 rpm and 3 °C. The supernatant was separated and 1000 µL was transferred into an HPLC vial. For the sugar analysis, a Shimadzu HPLC-set (USA Manufacturing Inc., Canby, OR, USA) was used, which consisted of two pumps LC-20AD, controller CBM-20A, column oven SIL-20AC, and spectrometer RID. The sugars were separated under isocratic conditions with a flow rate of 1 mL min$^{-1}$ using 80% acetone with deionized water. The total time of the analysis was 15 min. A Phenomenex Luna NH2 column was used for sugar compound analysis (glucose, fructose, and sucrose) with 99.9% pure standards from Sigma-Aldrich, Poland. The analysis times for the standards were considered (Chromatograms in Figure S2).

### 2.3.3. Vitamin C Analysis

To determine the amount of vitamin C in the plant powder, high-performance liquid chromatography (HPLC) coupled with a UV–VIS spectrometer from Shimadzu Manufacturing Inc. was used [10]. Firstly, one hundred milligrams of freeze-dried plant powder was weighed in a plastic tube. Then, five milliliters of 5% meta-phosphoric acid was added to the tube and the sample was shaken and extracted in an ultrasonic bath for 10 min at a temperature of 30 °C and a frequency of 5.5 kHz. After this, the sample was centrifuged for 10 min at 6000 rpm at 0 °C. The supernatant was transferred to an HPLC vial and 100 µL was used for analysis. The analysis was performed with the mobile phase acetic buffer (pH 4.4), which was composed of two solutions: 0.1 M acetic acid (glacial, 99.9% purity) and 0.1 M sodium acetate. The mobile phase was prepared with a ratio of 63:37 *v/v*. Isocratic flow was used with 1 mL min$^{-1}$. Four replicates were made for each analytical combination (Chromatograms in Figure S1).

### 2.3.4. Fibers, Ash, and Proteins Analysis

The fibers of freeze-dried fireweed leaves were analyzed according to the Association of Official Agricultural Chemists methods [11], ash content determined through combustion at 550 °C [12], and protein content was measured by the Kjeldahl method using KJELDATHERM (Gerhardt, Konigswinter, Germany) [13].

### 2.4. Statistical Analysis

The statistical analysis was conducted using Statgraphics Centurion 15.2.11.0 (Stat-Point Technologies, Inc., located in Warrenton, VA, USA). The tables present the average values of nine (*n* = 9) individual measurements for the production system, which includes biodynamic, organic, and natural methods, and two fermentation times (24 h and 48 h) in comparison to the non-fermented control. A two-way analysis of variance was conducted with Tukey's test. Any differences between the groups at the significance level of *p* < 0.05 were considered statistically significant. Additionally, each mean value presented in the tables is accompanied by the standard error (SE).

## 3. Results

*Chemical Composition of Fireweed Leaves*

The obtained results showed how fibers, ash, and proteins are distributed in organic fireweed leaves according to the method of growing and the duration of solid-phase fermentation (Table 1). The used variants of fermentation of the fireweed leaves did not significantly affect the quantities of fibers, ash, and proteins in the fireweed leaves. The highest content of fibers was in the organically grown fireweed leaves, while the highest quantities of ash and proteins were determined in the biodynamically grown fireweed leaves.

**Table 1.** The mean value for fibers, ash, and proteins (%, D.M.) of fireweed leaves based on various production systems and fermentation time (h).

| Production System | Fibers | Ash | Proteins |
|---|---|---|---|
| | %, D.M. | | |
| Biodynamic | 10.17 ± 0.21 [1] AB [2] | 6.52 ± 0.13 A | 18.15 ± 0.50 A |
| Organic | 10.37 ± 0.22 A | 4.66 ± 0.21 C | 15.89 ± 0.38 C |
| Natural | 9.91 ± 0.26 B | 5.36 ± 0.18 B | 17.15 ± 0.64 B |
| Control | 10.04 ± 0.14 a | 5.36 ± 0.90 a | 16.59 ± 0.97 a |
| 24 h | 10.04 ± 0.32 a | 5.51 ± 0.85 a | 17.07 ± 1.12 a |
| 48 h | 10.35 ± 0.29 a | 5.68 ± 0.79 a | 17.52 ± 1.08 a |
| *p*-Value | | | |
| Production systems | 0.0127 | <0.0001 | <0.0001 |
| Fermentation duration time (h) | N.S. [3] | N.S. | N.S. |

[1] The data are presented as the mean value with the standard error (SE) and ANOVA *p*-value. [2] Means in columns with the same letter are not significantly different at the 5% level (*p* < 0.05). [3] N.S.—not significant.

During the interaction, we noticed some changes in the fermentation process: it did not significantly change the quantities of fibers, ash, and proteins in fireweed leaves (Table 1). In the all-experimental combinations, we observed that the highest content of fibers was in the organically grown and 48 h fermented fireweed leaves, while the lowest quantity of fibers was in the naturally grown and 24 h fermented leaves samples.

The highest content of ash was determined in the biodynamically grown and 48 h fermented fireweed leaves, meanwhile, the lowest quantity of ash was determined in the organically grown and not-fermented fireweed leaves (control). The highest content of proteins was established in the biodynamic fireweed leaves, which were fermented for 48 h; on the contrary, the lowest content was in the organic not-fermented fireweed leaves. The natural treatment with 48 h of fermentation was similar to the biodynamic control sample (Table 2).

**Table 2.** The contents of fibers, ash, and proteins (%, D.M.) of fireweed leaves based on various production systems and fermentation time (h).

| Production System | Duration | Fibers | Ash | Proteins |
|---|---|---|---|---|
| | | %, D.M. | | |
| Biodynamic | Control | 10.08 ± 0.09 [1] bc [2] | 6.43 ± 0.01 a | 17.72 ± 0.09 ab |
| | 24 h | 10.05 ± 0.31 bc | 6.54 ± 0.08 a | 18.29 ± 0.09 a |
| | 48 h | 10.37 ± 0.02 ab | 6.60 ± 0.22 a | 18.41 ± 0.83 a |
| Organic | Control | 10.15 ± 0.04 bc | 4.44 ± 0.03 e | 15.58 ± 0.05 e |
| | 24 h | 10.34 ± 0.11 ab | 4.69 ± 0.02 de | 15.80 ± 0.24 de |
| | 48 h | 10.61 ± 0.12 a | 4.87 ± 0.17 d | 16.28 ± 0.40 de |
| Natural | Control | 9.89 ± 0.15 c | 5.22 ± 0.01 c | 16.47 ± 0.27 cd |
| | 24 h | 9.73 ± 0.17 c | 5.30 ± 0.14 bc | 17.11 ± 0.14 bc |
| | 48 h | 10.10 ± 0.39 bc | 5.55 ± 0.14 b | 17.86 ± 0.20 ab |
| *p*-Value | | | | |
| Production systems × fermentation duration time (h) | | N.S. | N.S. [3] | N.S. |

[1] The data are presented as the mean value with the standard error (SE) and ANOVA *p*-value. [2] Means in columns with the same letter are not significantly different at the 5% level (*p* < 0.05). [3] N.S.—not significant.

The obtained results showed that there were no statistical differences in total sugars as well as fructose and glucose content among the experimental combinations: organic, biodynamic, and natural. Only saccharose concentration was significantly higher in the natural samples compared to the organic and biodynamic samples. Fermentation processing diminishes the level of total sugars and all sugar fractions. We observed that after 24 h of fermentation, the level of all sugars diminished, compared to the control combination. Long fermentation contributed to a further decrease in the content of total sugars and all component sugars (fructose, glucose). In the case of sucrose, this compound was not detected after 48 h of fermentation of fireweed leaves (Table 3).

**Table 3.** The content of individually identified sugars (mg 100 g$^{-1}$ D.M.) in fireweed leaves based on various production systems and fermentation time (h).

| Production System | Total Sugars | Fructose | Glucose | Sucrose |
|---|---|---|---|---|
| Biodynamic | 5.86 ± 0.4 [1] A [2] | 2.01 ± 0.1 A | 3.30 ± 0.2 A | 0.55 ± 0.1 B |
| Organic | 5.95 ± 0.3 A | 1.98 ± 0.1 A | 3.42 ± 0.1 A | 0.55 ± 0.1 B |
| Natural | 5.86 ± 0.5 A | 1.99 ± 0.1 A | 3.30 ± 0.3 A | 0.57 ± 0.1 A |
| Control | 7.08 ± 0.05 a | 2.30 ± 0.01 a | 3.89 ± 0.04 a | 0.89 ± 0.01 a |
| 24 h | 6.37 ± 0.07 b | 2.07 ± 0.03 b | 3.51 ± 0.06 b | 0.78 ± 0.01 b |
| 48 h | 4.23 ± 0.09 c | 1.61 ± 0.01 c | 2.62 ± 0.08 c | not detected |
| *p*-Value | | | | |
| Production systems | N.S. [3] | N.S. | N.S. | 0.0001 |
| Fermentation duration time (h) | <0.0001 | <0.0001 | <0.0001 | <0.0001 |

[1] The data are presented as the mean value with the standard error (SE) and ANOVA *p*-value. [2] Means in columns with the same letter are not significantly different at the 5% level (*p* < 0.05). [3] N.S.—not significant.

In the case of total sugars, we observed that short fermentation decreased sugar concentration. The highest decrease in total sugar content was observed in the biodynamic samples (12.8%), and the smallest in the natural samples (7.5%), compared to the unfermented control. Consecutive hours of fermentation are not recommended, especially for samples from the natural combination. It was observed that these samples lost the highest total sugars compared to the short fermented (24 h) fireweed leaves. A similar situation was observed with the concentration of fructose and glucose. It seems that only short fermentation duration is preferable for all experimental combinations, even though a decrease in total sugars was observed after 24 h compared to the unfermented control. It is worth pointing out that only short fermentation is recommended for experimental samples. After 48 h of fermentation of fireweed leaves, sucrose was not detected (Table 4).

The naturally produced fireweeds were characterized by a statistically significant concentration of vitamin C. In the organic samples, we observed the lowest concentration of vitamin C in the leaves. It seems that the fermentation process stimulated the concentration of vitamin C in the experimental samples. After 48 h of duration, we observed an almost double increase in the vitamin C concentration compared to the not-fermented samples. In the case of dehydroascorbic and L-ascorbic acid fractions, a significant concentration was observed in the organic samples as well. On the other hand, in the case of dehydroascorbic acid (DHA), the natural samples were characterized by a significantly higher content of the vitamin C fraction. In the case of L-ascorbic acid, these were organic samples. A longer fermentation process is positive in the context of an increase in the content of both components of vitamin C. In both cases, after 48 h, a significantly higher concentration of the tested compounds was observed (Table 5).

**Table 4.** The content of individually identified sugars (mg 100 g$^{-1}$ D.M.) in fireweed leaves based on various production systems and fermentation time (h).

| Production Systems | Fermentation Duration Time | Total Sugars | Fructose | Glucose | Sucrose |
|---|---|---|---|---|---|
| Biodynamic | Control | 7.15 ± 0.05 [1] a [2] | 2.32 ± 0.03 a | 3.94 ± 0.04 a | 0.88 ± 0.01 a |
| | 24 h | 6.23 ± 0.04 b | 2.14 ± 0.01 b | 3.31 ± 0.04 b | 0.77 ± 0.01 b |
| | 48 h | 4.21 ± 0.02 c | 1.56 ± 0.03 c | 2.65 ± 0.01 c | not detected |
| Organic | Control | 6.91 ± 0.06 a | 2.26 ± 0.01 a | 3.75 ± 0.05 a | 0.89 ± 0.01 a |
| | 24 h | 6.22 ± 0.04 b | 1.97 ± 0.04 b | 3.49 ± 0.03 b | 0.76 ± 0.01 b |
| | 48 h | 4.73 ± 0.15 c | 1.72 ± 0.04 c | 3.01 ± 0.12 c | not detected |
| Natural | Control | 7.19 ± 0.02 a | 2.32 ± 0.01 a | 3.97 ± 0.02 a | 0.90 ± 0.01 a |
| | 24 h | 6.65 ± 0.05 b | 2.11 ± 0.04 b | 3.74 ± 0.01 b | 0.81 ± 0.01 b |
| | 48 h | 3.74 ± 0.08 c | 1.55 ± 0.02 c | 2.19 ± 0.09 | not detected |
| *p*-Value | | | | | |
| Interaction | | <0.0001 | 0.0003 | <0.0001 | 0.0002 |

[1] The data are presented as the mean value with the standard error (SE) and ANOVA *p*-value. [2] Means in columns with the same letter are not significantly different at the 5% level ($p < 0.05$).

**Table 5.** The mean value for vitamin C and its fractions (mg 100 g$^{-1}$ D.M.) in fireweed leaves based on various production systems and fermentation time (h).

| Production Systems | Vitamin C | Dehydroascorbic Acid | L-ascorbic Acid |
|---|---|---|---|
| Biodynamic | 261.35 ± 14.0 [1] C [2] | 123.90 ± 10.3 C | 137.45 ± 4.1 A |
| Organic | 515.29 ± 71.7 A | 411.67 ± 61.5 B | 103.61 ± 11.7 B |
| Natural | 447.06 ± 38.1 B | 382.42 ± 34.0 A | 64.64 ± 6.0 C |
| Control | 247.19 ± 12.8 c | 166.14 ± 20.9 c | 81.05 ± 10.9 c |
| 24 h | 441.80 ± 50.8 b | 347.86 ± 58.4 b | 93.94 ± 11.2 b |
| 48 h | 534.70 ± 40.8 a | 403.99 ± 51.1 a | 130.71 ± 10.5 a |
| *p*-Value | | | |
| Production systems | <0.0001 | <0.0001 | <0.0001 |
| Fermentation duration time (h) | <0.0001 | <0.0001 | <0.0001 |

[1] The data are presented as the mean value with the standard error (SE) and ANOVA *p*-value. [2] Means in columns with the same letter are not significantly different at the 5% level ($p < 0.05$).

The highest increase in the sum of vitamin C as well as both fractions was observed in the organic samples, especially after 24 h of fermentation: vitamin C (+171%, DHA +223%, and l-ascorbic acid +45%) compared to the control not-fermented samples. The next 48 h of fermentation showed progress in increasing vitamin C and both fractions. Taking into account the two factors and their interaction, it can be presumed that after 48 h from fermentation, there was a further increase in the content of vitamin C in the tested samples. The greatest increase was observed in the biodynamic samples (+33%) compared to the samples after short fermentation. Organic samples gained the most DHA after 48 h (+58%). In the case of the natural samples, this increase was also observed after 48 h (+70%) compared to the samples after 24 h of fermentation (Table 6).

**Table 6.** The content of vitamin C and its fractions (mg 100 g$^{-1}$ D.M.) in fireweed leaves based on various production systems and fermentation time (h).

| Production Systems | Duration | Vitamin C | Dehydroascorbic Acid | L-ascorbic Acid |
|---|---|---|---|---|
| Biodynamic | Control | 224.74 ± 2.1 [1] e [2] | 98.39 ± 0.7 e | 126.35 ± 1.7 c |
| | 24 h | 239.79 ± 4.2 e | 105.89 ± 1.8 e | 133.90 ± 4.6 b |
| | 48 h | 319.51 ± 3.5 d | 167.41 ± 0.5 d | 152.10 ± 2.9 a |
| Organic | Control | 221.69 ± 4.7 e | 156.29 ± 6.2 d | 65.40 ± 1.8 e |
| | 24 h | 601.51 ± 21.4 ab | 506.26 ± 21.2 b | 95.24 ± 1.4 d |
| | 48 h | 722.67 ± 13.3 a | 572.48 ± 13.6 a | 150.19 ± 0.3 a |
| Natural | Control | 295.15 ± 17.2 e | 243.75 ± 17.9 c | 51.39 ± 0.7 f |
| | 24 h | 484.12 ± 5.3 c | 431.43 ± 5.3 b | 52.68 ± 0.2 f |
| | 48 h | 561.91 ± 14.2 b | 472.07 ± 13.1 b | 89.83 ± 1.5 d |
| *p*-Value | | | | |
| Production systems × fermentation duration time (h) | | <0.0001 | <0.0001 | <0.0001 |

[1] The data are presented as the mean value with the standard error (SE) and ANOVA *p*-value. [2] Means in columns with the same letter are not significantly different at the 5% level ($p < 0.05$).

## 4. Discussion

During the solid-phase fermentation of fireweed leaves, compounds that are composed of multiple molecules are broken down into simpler ones and some substances transform, resulting in qualitative and quantitative changes. The use of fermentation as an extraction method can increase the content of bioactive compounds, and also induce chemical changes in the original molecules [14–16].

In the case of interactions between samples, during our observation, we noticed the process of fermentation did not significantly change the quantities of fibers, ash, and proteins in the fireweed leaves. However, in the biodynamic and organic growing systems, the quantities of fibers were higher, accordingly: 1.88% and 2.56%, compared with the natural growing fireweeds. The highest amounts of ash and proteins were found in the biodynamic fireweed leaves. The same quantities of proteins were found in the natural fireweed at 48 h of fermentation and in the biodynamic control treatment (Table 2).

Sugars, organic acids, and amino acids are the primary sources of energy in plants, produced through photosynthesis and respiration [17]. During solid-phase fermentation, one of the crucial steps is the hydrolysis of primary polymeric substances into finer substances such as disaccharides (e.g., sucrose) that are converted into monosaccharides (e.g., fructose and glucose). The presented data show that, during aerobic fermentation, the total amount of sugars, including fructose, glucose, and sucrose, decreased significantly. This can be explained by the fact that disaccharide sucrose is broken down into monosaccharides during solid-phase fermentation. However, the different growing systems did not significantly affect the total sugar content (Table 4).

Vitamin C is an essential nutrient for maintaining human health. Numerous diseases, including infections, cancer, cardiovascular disease, stroke, diabetes, and sepsis, have been linked to poor levels of vitamin C [18,19]. Reduced vitamin C levels during illness are often attributed to a combination of high turnover caused by oxidative stress and inflammation, as well as a decreased intake of vitamin C associated with the diseases [20,21].

From the presented data (Table 6), we can see that solid-phase fermentation significantly increased the amounts of vitamin C in its oxidized form, dehydroascorbic acid, and L-ascorbic acid. Vitamin C's highest content was found in the organic 48 h fermented fireweed leaves (722.67 mg 100 g$^{-1}$ D.M.). During fermentation, the leaves are crushed and pressed, which enhances the degradation processes of the cell walls and improves the diffusion of biologically active compounds from the inner parts of the cells. This results in a more efficient process of extracting compounds. There are some data suggesting that certain solid-phase fermentation parameters can activate the accumulation process of

certain bioactive compounds in fireweed leaves [22,23]. Different growing systems did not have a significant influence on vitamin C (Table 6).

## 5. Conclusions

Based on the results of the experiment, different growing systems and solid-phase fermentation affected the amounts of different substances in the leaves of fireweeds. The highest content of fibers was in the organically grown fireweed leaves, while the highest quantities of ash were measured in the biodynamically grown fireweed leaves. The highest quantities of proteins were in the biodynamic samples (24 and 48 h of fermentation), but in the naturally growing fireweeds (48 h of fermentation), protein quantities were similar to the biodynamic control sample.

The obtained results showed no statistical differences in total sugars and fructose and glucose content among the experimental combinations: organic, biodynamic, and natural. Only saccharose concentration was significantly higher in the natural samples compared to the organic and biodynamic samples. Fermentation processing diminishes the level of total sugars and all sugar fractions. It is worth pointing out that only short fermentation could be used for the experimental samples.

The naturally produced fireweeds were characterized by a statistically significant concentration of vitamin C. In the organic samples, we observed the lowest concentration of vitamin C in the leaves. It seems that the fermentation process stimulated the concentration of vitamin C in the experimental samples.

Based on the available data, it appears feasible to provide a recommendation: 48 h of fermented fireweed leaves, grown organically, as a source of vitamin C and fibers, and leaves, grown biodynamically, as a source of ash and proteins, and naturally grown, not fermented, leaves for sugars.

In sum, solid-phase fermentation has been found to lower the sugar content in fireweed leaves. This study's findings suggest that an organic growing system could be recommended for the food and pharmacy industry to produce fireweed leaves as functional food products with higher vitamin C. Additionally, biodynamically grown fireweed leaves could be highly beneficial for people due to their high ash and protein content, while naturally grown leaves would be ideal for those looking for sugar.

**Supplementary Materials:** The following supporting information can be downloaded at: https://www.mdpi.com/article/10.3390/horticulturae9111245/s1, Figure S1: Chromatograms from identification of vitamin C in fireweed: (A) organic, (B) natural, (C) bioynamic leaves: [1] dehydroascorbic acid, [2] l-ascorbic acid. Figure S2: Chromatograms from identification of sugars in fireweed: (A) organic, (B) natural, (C) biodynamic leaves: [1] fructose, [2] glucose, [3] sucrose.

**Author Contributions:** Conceptualization, M.L.; methodology, E.H.; project administration E.J.; resources M.L. and S.T.; software E.H. and A.V.; formal analysis N.V.; validation E.H., S.T. and E.J.; visualization M.L. and A.V.; writing—original draft, M.L. and N.V.; writing—review and editing J.K. and E.J.; supervision E.J. and J.K.; data curation, M.L. All authors have read and agreed to the published version of the manuscript.

**Funding:** The study was funded by the Ekhagastiftelsen for application "Studies of the variability of biologically active and anticancer compounds in organically and biodynamically grown and fermented fireweed leaves" (No. 2021-67).

**Data Availability Statement:** Data are contained within the article.

**Conflicts of Interest:** The authors declare no conflict of interest.

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
