# Peer review of "The Variability of Proximate Composition, Sugars, and Vitamin C in Natural, Organic, and Biodynamic, and Fermented Leaves of Fireweed (Chamerion angustifolium (L.) Holub (Onagraceae))"

_horticulturae, doi:10.3390/horticulturae9111245_

Round 1
Reviewer 1 Report
Comments and Suggestions for Authors
- The current scientific name of fireweed should be incorporated in the title: Chamerion angustifolium (L.) Holub (Onagraceae).
- lines 19 and 20, paragraph "...as a way to improve one's health while also enhancing one's diet." I think it is better to say "...as a way to improve health and at the same time improve diet."
Author Response
Replies to Reviewers’ comments
Ms. Ref. No.: horticulturae-2684443, Title: “Studies of the variability of proximate composition, sugars, and vitamin C in naturally, organically, and biodynamically grown and fermented fireweed (Chamerion angustifolium (L.) Holub) leaves”
Journal: Applied Sciences
Dear Editor,
I’m pleased to submit the revised manuscript after addressing all Reviewers’ comments. All adopted changes are marked in the manuscript with the blue background. Below you can find our replies to each of the Reviewers’ comments/suggestions/questions.
Reviewer no. 1:
Thank you very much for the review and for your positive recommendation to publish our manuscript in the “Horticulturae” journal.
Comment 1: “The current scientific name of fireweed should be incorporated in the title: Chamerion angustifolium (L.) Holub (Onagraceae)“
Authors’ response: According to Your suggestion, we have added new information to the title.
Comment 2: “Lines 19 and 20, paragraph "...as a way to improve one's health while also enhancing one's diet." I think it is better to say "...as a way to improve health and at the same time improve diet."
Authors’ response: According to Your suggestion, we have added Your sentence.
Please see the attachment.

Reviewer 2 Report
Comments and Suggestions for Authors
The work is interesting, the qualitative and quantitative changes in fireweed leaves due to the production system and fermentitive process times reported in this study.
However, the authors are required to clarify and correct some points in the manuscript.
1. In the materials and methods section, lines 91-92, Indicate, what is the chemical (and biological if applicable) composition of the compost and biodynamic compost?
2. It is necessary to describe in detail the BD preparation 500 (line 93-94). Also describe in detail the BD preparation 501 (line 95). What is the difference between these two preparations?
3. Separate the number from the units (lines 125, 130, 131).
4. The "-1" must be as a superscript (line 135).
5. In the results section. Review all tables in the manuscript, the instructions for authors should be reviewed to properly format the tables.
6. The titles of the first column of tables 1, 3, and 5 are not correct, please review them.
7. In tables 1, 3, and 5 there is a mixture of upper and lower case letters in the comparison of means, please correct.
8. In lines 187-189 it is written "The highest content of proteins was established in biodynamic fireweed leaves, which were fermented for 48 hours, on the contrary, the lowest content was in organic not fermented fireweed leaves...". However, it is not entirely correct, the control natural treatment with 48 h of fermentation is similar to the biodynamic one. This statement needs to be corrected. Is not the same tendency for ash.
9. The discussion and conclusions must be in accordance with the possible changes made according to the corrections.

Author Response
Replies to Reviewers’ comments
Ms. Ref. No.: horticulturae-2684443, Title: “Studies of the variability of proximate composition, sugars, and vitamin C in naturally, organically, and biodynamically grown and fermented fireweed (Chamerion angustifolium (L.) Holub) leaves”
Journal: Applied Sciences
Dear Editor,
I’m pleased to submit the revised manuscript after addressing all Reviewers’ comments. All adopted changes are marked in the manuscript with the green background. Below you can find our replies to each of the Reviewers’ comments/suggestions/questions.
Reviewer no. 2:
Thank you very much for the review and for your positive recommendation to publish our manuscript in the “Horticulturae” journal.
Comment 1: “In the materials and methods section, lines 91-92, Indicate, what is the chemical (and biological if applicable) composition of the compost and biodynamic compost?”
Authors’ response: We have added additional information. Lines 91-94.
Comment 2: “It is necessary to describe in detail the BD preparation 500 (line 93-94). Also describe in detail the BD preparation 501 (line 95). What is the difference between these two preparations?”
Authors’ response: We have added additional information. Lines 97-104.
Comment 3: “Separate the number from the units (lines 125, 130, 131).”
Authors’ response: We have corrected it.
Comment 4: “The "-1" must be as a superscript (line 135).”
Authors’ response: We have corrected it.
Comment 5: “In the results section. Review all tables in the manuscript, the instructions for authors should be reviewed to properly format the tables.”
Authors’ response: We have corrected it.
Comment 6: “The titles of the first column of tables 1, 3, and 5 are not correct, please review them.”
Authors’ response: We have corrected it.
Comment 7: “In tables 1, 3, and 5 there is a mixture of upper and lower case letters in the comparison of means, please correct.”
Authors’ response: Upper case letters are for production systems (natural, organic, biodynamic): A, B, C, and lower case: a, b, c letters are for fermentation time (control, 24 h, 48 h) – so it is correct.
Comment 8: “In lines 187-189 it is written "The highest content of proteins was established in biodynamic fireweed leaves, which were fermented for 48 hours, on the contrary, the lowest content was in organic not fermented fireweed leaves...". However, it is not entirely correct, the control natural treatment with 48 h of fermentation is similar to the biodynamic one. This statement needs to be corrected. Is not the same tendency for ash.”
Authors’ response: According to Your advice, we have corrected it. Lines 192-193.
Comment 9: “The discussion and conclusions must be in accordance with the possible changes made according to the corrections.”
Authors’ response: According to Your advice, we have corrected it. Lines 269-271, 301-304.
Please see the attachment.

Reviewer 3 Report
Comments and Suggestions for Authors
The article Studies of the variability of proximate composition, sugars, and vitamin C in natural, organic, and biodynamic, and fermented leaves of fireweed explores an interesting topic but has serious flaws that must be corrected before consideration for publication of the paper.
1. In the Introduction part the authors should specify the terms organic and biodynamic grown conditions (Lines 53-57)
2. The plant itself should be more appropriately described – the Latin name must be provided, as well as the biochemical composition of the plant and its pharmacological mechanisms of activities.
3. The term solid phase fermentation technology must be better defined (Lines 64-66)
4. If the aim of the study was to compare the impact of different types of farming on fireweed leaf quality in the context of the firewood leaves medicinal usage, then more appropriate bioactive substances must be quantified – e.g., flavonoids, phenolic acids, etc. The quantified compounds are not relevant for pharmacological activity (with the exception of vitamin C).
5. The text formatting is not uniform (line spacing)
6. Please add the part Chemicals with listed chemicals used in this research.
7. In the Experiment part there is no data on the treatment of organically grown plants.
8. Please provide references for fiber, ash, and protein analysis, as well as a short description of the procedure.
9. There is no data on HPLC procedure validation for sugars and vitamin C, nor a suitable reference is provided.
10. There are no chromatograms provided within the manuscript.
11. Please revise the Tables. – letters a,b,c… should be in superscript.
12. Why is the content of sugars relevant for this study?
13. The discussion part should be better elaborated and supported by other studies of growing systems/solid phase extraction impact on bioactive compounds contents.
Comments on the Quality of English LanguageMinor editing required
Author Response
Replies to Reviewers’ comments
Ms. Ref. No.: horticulturae-2684443, Title: “Studies of the variability of proximate composition, sugars, and vitamin C in naturally, organically, and biodynamically grown and fermented fireweed (Chamerion angustifolium (L.) Holub) leaves”
Journal: Applied Sciences
Dear Editor,
I’m pleased to submit the revised manuscript after addressing all Reviewers’ comments. All adopted changes are marked in the manuscript with the yellow background. Below you can find our replies to each of the Reviewers’ comments/suggestions/questions.
Reviewer no. 1:
Thank you very much for the review and for your positive recommendation to publish our manuscript in the “Horticulturae” journal.
Comment 1: “In the Introduction part the authors should specify the terms organic and biodynamic grown conditions (Lines 53-57).”
Authors’ response: We have added new information about organic and biodynamic in the materials and methods (Lines 103-125).
Comment 2: “The plant itself should be more appropriately described – the Latin name must be provided, as well as the biochemical composition of the plant and its pharmacological mechanisms of activities.”
Authors’ response: According to Your suggestion we have added more information (Lines 59-66).
Comment 3. “The term solid phase fermentation technology must be better defined (Lines 64-66).”
Authors’ response: According to Your suggestion we have added more information about solid phase fermentation (Lines 71-80).
Comment 4. “If the aim of the study was to compare the impact of different types of farming on fireweed leaf quality in the context of the firewood leaves medicinal usage, then more appropriate bioactive substances must be quantified – e.g., flavonoids, phenolic acids, etc. The quantified compounds are not relevant for pharmacological activity (with the exception of vitamin C).”
Authors’ response: We clarify that the aim was to compare the impact of different types of farming on fireweed leaf quality in the context of the fireweed leaves that could be used not only for medicinal properties, but also for functional food, or animal feed, or another purpose. Of course, we have done a lot of experiments with fireweed biologically active substances: polyphenols, carotenoids, chlorophylls, and others. However this article is concentrated on proximate constituents, sugars, and vitamin C.
Comment 5. “The text formatting is not uniform (line spacing).”
Authors’ response: We have corrected it.
Comment 6. “Please add the part Chemicals with listed chemicals used in this research.”
Authors’ response: According to Your suggestion we have added information about chemicals (Lines 154-157).
Comment 7. “In the Experiment part there is no data on the treatment of organically grown plants.”
Authors’ response: We have added more information about organic growing (Lines 102-124).
Comment 8. “Please provide references for fiber, ash, and protein analysis, as well as a short description of the procedure.”
Authors’ response: According to Your suggestion we have added references (Lines 186-189).
Comment 9. “There is no data on HPLC procedure validation for sugars and vitamin C, nor a suitable reference is provided.”
Authors’ response: We have added references for method validation:
Ponder A., Hallmann E. The nutritional value and vitamin C content of different raspberry cultivars from organic and conventional production, Journal of Food Composition and Analysis, 2020, 87, 1-14.
Comment 10. “There are no chromatograms provided within the manuscript.”
Authors’ response: According to Your suggestion we have added chromatograms as Figures 1S and 2S for supplementary material.
Comment 11. “Please revise the Tables. – letters a,b,c… should be in superscript.”
Authors’ response: We use temple horticulturae-template.dot and we don’t see this that a, b, c must be in superscript.
Comment 12. “Why is the content of sugars relevant for this study?”
Authors’ response: In this article we wanted to show the tendency of sugars during different growing systems and different fermentation conditions.
Comment 13. “The discussion part should be better elaborated and supported by other studies of growing systems/solid phase extraction impact on bioactive compounds contents.”
Authors’ response: There is not much information about biodynamic system and about solid-phase fermentation with fireweed leaves, so it is difficult to discuss a lot. But maybe in the future more studies will be done and then we could discuss more about it.
Please see the attachment.

Round 2
Reviewer 2 Report
Comments and Suggestions for Authors
It is an interesting research work. From my point of view, the corrections made by the authors substantially improved the presentation of the manuscript.
Author Response
Thank You very much!
Reviewer 3 Report
Comments and Suggestions for Authors
The "major revision" is due to the lack of supplementary material 1S and 2S which are mentioned in the author's response (answer to comment 10).
Please provide the chromatograms.
Author Response
I added the chromatograms, but maybe it was a technical mistake. I will do it once more now.
Round 3
Reviewer 3 Report
Comments and Suggestions for Authors
After the author's response and upload of supplementary materials, I recommend the paper be published.
Author Response
Thank You very much!